Mammal dung–dung beetle trophic networks: an improved method based on gut-content DNA

Pedersen Karen M. karen.pedersen@tu-darmstadt.de
von Beeren Christoph
Oggioni Arianna
Blüthgen Nico
Biology, Technical University of Darmstadt , Darmstadt , Hessen , Germany
Kent Clement
Electronic publication date: 2024 Mar 15
Publication date: 2024
Volume: 12
Electronic Location ID: e16627
Received 2023 Jan 7; Accepted 2023 Nov 16
Copyright: ©2024 Pedersen et al.
Copyright year: 2024
Copyright holder: Pedersen et al.
License: This is an open access article distributed under the terms of the Creative Commons Attribution License, which permits unrestricted use, distribution, reproduction and adaptation in any medium and for any purpose provided that it is properly attributed. For attribution, the original author(s), title, publication source (PeerJ) and either DOI or URL of the article must be cited.
License URL: https://creativecommons.org/licenses/by/4.0/

Keywords: Dung beetle, Tropic network, iDNA, Dung beetle diet, Mammal-dung beetle, Gut content, Gut content dna, Mammal-dung beetle networks

Funding: Deutsche Forschungsgemeinschaft (DFG) in the framework of the collaborative Research Unit FOR 5207 REASSEMBLY subproject BL 960/13 This work was supported by the Deutsche Forschungsgemeinschaft (DFG) in the framework of the collaborative Research Unit FOR 5207 REASSEMBLY (subproject BL 960/13). The funders had no role in study design, data collection and analysis, decision to publish, or preparation of the manuscript.

==============================
Background

Dung beetles provide many important ecosystem services, including dung decomposition, pathogen control, soil aeration, and secondary seed dispersal. Yet, the biology of most dung beetles remains unknown. Natural diets are poorly studied, partly because previous research has focused on choice or attraction experiments using few, easily accessible dung types from zoo animals, farm animals, or humans. This way, many links within natural food webs have certainly been missed. In this work, we aimed to establish a protocol to analyze the natural diets of dung beetles using DNA gut barcoding.

Methods

First, the feasibility of gut-content DNA extraction and amplification of 12s rDNA from six different mammal dung types was tested in the laboratory. We then applied the method to beetles caught in pitfall traps in Ecuador and Germany by using 12s rDNA primers. For a subset of the dung beetles caught in the Ecuador sampling, we also used 16s rDNA primers to see if these would improve the number of species we could identify. We predicted the likelihood of amplifying DNA using gut fullness, DNA concentration, PCR primer, collection method, and beetle species as predictor variables in a dominance analysis. Based on the gut barcodes, we generated a dung beetle-mammal network for both field sites (Ecuador and Germany) and analyzed the levels of network specificity.

Results

We successfully amplified mammal DNA from dung beetle gut contents for 128 specimens, which included such prominent species as Panthera onca (jaguar) and Puma concolor (puma). The overall success rate of DNA amplification was 53%. The best predictors for amplification success were gut fullness and DNA concentration, suggesting the success rate can be increased by focusing on beetles with a full gut. The mammal dung–dung beetle networks differed from purely random network models and showed a moderate degree of network specialization (H2′: Ecuador = 0.49; Germany = 0.41).

Conclusion

We here present a reliable method of extracting and amplifying gut-content DNA from dung beetles. Identifying mammal dung via DNA reference libraries, we created mammal dung-dung beetle trophic networks. This has benefits over previous methods because we inventoried the natural mammal dung resources of dung beetles instead of using artificial mammal baits. Our results revealed higher levels of specialization than expected and more rodent DNA than expected in Germany, suggesting that the presented method provides more detailed insights into mammal dung–dung beetle networks. In addition, the method could have applications for mammal monitoring in many ecosystems.

Introduction

Dung beetles have been used as indicator species of habitat quality because of their sensitivity to habitat degradation, both in terms of deforestation and defaunation and their ecological importance (Nichols et al., 2007; Korasaki et al., 2013; Bicknell et al., 2014; Ong, Slade & Lim, 2020). The primary source of nutrition for most dung beetles is thought to be moist dung from large mammals (Hanski & Cambefort, 1991; Holter & Scholtz Clarke, 2007; Raine et al., 2018; Raine & Slade, 2019). However, some species have been documented consuming rodent dung, carrion, bird dung, millipedes, or rotten fruit (Schmitt, Krell & Linsenmair, 2004; Larsen, Williams & Kremen, 2005; Halffter & Halffter, 2009; Kerley et al., 2018; Silva, Vaz-de Mello & Barclay, 2018). Adult dung beetles have soft mouthparts that do not allow them to chew hard parts like bits of grass, often present in large herbivores’ dung. The larva, however, possess chewing mouthparts and are hypothesized to be able to exploit these solid parts within the dung (Halffter & Edmonds, 1982).

By feeding on mammal dung, rotting carrion, and fruits, dung beetles provide essential ecosystem services such as soil nutrient recycling, soil aeration, pathogen control, and secondary seed dispersal (Hanski & Cambefort, 1991; Nichols et al., 2007). However, the dietary niches of most dung beetles remain unknown, and those that have been described are primarily based on compilations of observations rather than quantitative data (Young, 1981; Hanski & Cambefort, 1991; Nichols et al., 2007; Edmonds & Zidek, 2010). For instance, there are some direct feeding observations at dung piles, but this is likely biased towards larger, more obvious dung, such as that from elephants, cows, and humans (Young, 1981; Hanski & Cambefort, 1991; Scholtz, Davis & Kryger, 2009). Further, experimentally deployed dung often represents common or readily available species, even using non-native species from zoos (Hanski & Cambefort, 1991; Frank et al., 2018a; Raine & Slade, 2019; Chiew et al., 2022). Overall, this has often led to the assumption that dung beetles primarily consume dung from large- and medium-sized mammals (Hanski & Cambefort, 1991; Scholtz, Davis & Kryger, 2009; Simmons & Ridsdill-Smith, 2011; Bogoni & Hernández, 2014; Frank et al., 2018a; Frank et al., 2018b; Raine et al., 2018; Raine & Slade, 2019; Bogoni, Da Silva & Peres, 2019).

This approach has then been passed on to the realm of mammal dung-dung beetle networks (Frank et al., 2018a; Raine & Slade, 2019; Chiew et al., 2022; Pryke, Roets & Samways, 2022). For example, rodent dung is often excluded from cafeteria-style experiments or other dung attraction experiments designed to study mammal dung-dung beetle networks (Bogoni & Hernández, 2014; Frank et al., 2018a; Raine & Slade, 2019; Ong, Slade & Lim, 2020; Chiew et al., 2022; Pryke, Roets & Samways, 2022). Traditional methods of creating mammal dung–dung beetle networks might thus be biased. The use of molecular techniques such as DNA gut barcoding and metabarcoding promises to uncover otherwise hidden trophic interactions (Wallinger et al., 2015; Hoenle et al., 2019; Avanesyan, Sutton & Lamp, 2021). Extracting mammal DNA from the beetles’ digestive tracts provides snapshots of the beetles’ last meals, thus allowing natural diets to be uncovered. A proof of concept was provided by Gómez & Kolokotronis (2017), who detected horse DNA in the guts of dung beetles collected directly on horse dung. Furthermore, the excrements of the dung beetle Circellium bacchus were used to identify its diet based on DNA metabarcodes (Kerley et al., 2018). Recently, a broader assessment of gut content DNA from 31 dung beetles in Borneo suggested that gut barcoding of dung beetles could be used to monitor mammals (Drinkwater et al., 2021).

The herein presented method shows similarities to the method presented by Drinkwater et al. (2021), with a few key differences. We used a second primer pair, Sanger sequencing, and a more aggressive washing protocol to minimize contaminations. The present work aims to develop a reliable, broadly applicable, cost-effective method to identify mammal species in dung beetle guts and better understand dung beetle diets. As a test case, we studied two mammal dung–dung beetle communities, one in a German temperate forest and another in an Ecuadorian tropical rainforest. By combining mammal identification via DNA barcoding with ecological network analysis, we unveiled the dung beetles’ diets and their levels of dietary specialization for both communities.

Materials & Methods

Beetle collection

We collected dung beetles from a temperate forest in Germany (49°51′54.19″N, 8°41′28.50″E) and a lowland tropical rainforest, as well as five pastures in Ecuador (0°30′20.52″N, 79°10′31.95″W Ecuador). Collections were made in September 2019 (Germany) and from January 2019 to June 2019 (Ecuador). Beetle collections were made under the Escuela Politecnica Nacional, Contrato Marco MAE-DNB-CM-2016-0068, and transported to Germany under beetle export authorization number 62-2019-EXP-CM-FAUDNBIMA. Beetles were collected using pitfall traps, baited with cow dung in Germany (seven pitfall traps, 41 beetles), and human dung in Ecuador (eight pitfall traps, 94 beetles). Beetles were also collected opportunistically by hand in Ecuador (N = 41).

Both cow and human dung are common attractants for dung beetle pitfall trapping (Hanski & Cambefort, 1991; Whipple & Hoback, 2012; Marsh et al., 2013; Frank et al., 2018a; Frank et al., 2018b). However, in preliminary trials, we realized that, in the Ecuadorian population, cow was less effective in attracting dung beetles than human dung. The latter attracted a higher number and diversity of dung beetles, so we decided to use human dung in Ecuador. We did not see a substantial difference between bait types in Germany and, therefore, decided to use cow dung as it is easier to handle. In all experiments, we euthanized beetle specimens quickly using either absolute ethanol or freezing to reduce the suffering of the specimens. Disposable nitrile gloves (VWR) were worn for all dung manipulations for personal protection, especially as human dung was used as a bait. After the baiting periods, human dung was disposed of in the same hole created for the pitfall trap following the Leave No Trace Principles for human waste disposal (www.lnt.org).

To minimize the contamination of beetle guts with cow or human dung, direct physical contact from dung beetles to baits was minimized. The pitfall traps consisted of a plastic cup leveled with the soil, a rain cover, and an overhanging tea bag with bait inside, mostly prohibiting unintended bait consumption (Frank et al., 2018a). Pitfall traps were installed 24 h before collection. When pitfall traps were collected, only dung beetles were collected. All other animals were released. After beetle collection in the field, specimens were either preserved in ethanol and then frozen (Ecuador) or simply frozen (Germany). Our field site in Ecuador is subject to occasional power outages and thus requires a second level of DNA preservation. Dung beetles were then identified morphologically by KMP using the latest species keys (Edmonds, 2000; Solís & Kohlmann, 2002; Edmonds & Zídek, 2004; Edmonds & Zídek, 2012; Vaz-De-Mello et al., 2011; Chamorro et al., 2018; Nunes, Nunes & Vaz-de Mello, 2018), and the reference collections of the Pontificia Universidad Católica (Ecuador) and the Escuela Superior Politécnica del Litoral (Ecuador).

Beetle dissection

After frequently amplifying human DNA in preliminary experiments before this study began, we developed a washing protocol in sterile conditions to minimize human DNA contamination, which markedly reduced the amplification of contaminants such as human DNA and prevented cross-contamination of samples. First, surfaces and tools were UV sterilized. Beetles were placed singly in distilled water and shaken for 30 s to remove dirt and external DNA from the outside. We transferred the beetles in a 2% NaClO (chlorine) solution under a fume hood where the air was constantly UV sterilized. The chlorine solution was washed off the beetles with 70% ethanol. Then, beetles were transferred to a dissection tray and dissected using various dissection tools. Tools were flamed, washed in 2% NaClO, rinsed in 70% EtOH, and then flamed again and cooled before each dissection. Cleaning solutions were changed every two beetles to prevent cross contamination and reduce waste liquid byproducts. After every beetle, the dissection tray was washed with 2% NaClO solution and 70% EtOH. We dissected 177 beetles, 135 from Ecuador and 42 from Germany.

For large beetles (body length > 1 cm), the digestive tract was removed and placed in the DNA extraction buffer of a Qiagen Blood and Tissue Kit (Qiagen, Hilden, Germany). For smaller beetles (body length < 1 cm), the entire abdomen was placed in the same DNA extraction buffer without further dissection. We used the entire abdomen for small specimens to prevent possible contamination. Their small size made it much harder to dissect, and as our first attempts at sequencing this way worked, we kept using the method. For a subset of 51 large beetles, we visually categorized gut fullness in the following way: (1) full—more than half the length of the intestines is full; (2) half-full—half or less than half the length of the intestine is full; (3) empty—there is no visible content in the intestine (Fig. S1).

DNA extraction and amplification

DNA was further purified using the Bio-RAD Micro Bio-Spin Columns P-30 Tris following the manufacturer’s instructions. This results in 75 µl of purified DNA, which can be used for PCR. We then ran PCRs using the Qiagen multiplex kit (with 1.5 µl of molecular grade water), 5 µl of Qiagen multiplex PCR Master Mix, then 0.5 µl of 10 µM for the forward and reverse primers, and with the addition of 0.5 µl of bovine serum albumin (BSA) to counteract remaining PCR inhibitors and 2 µl of template DNA. The PCRs started with an initial activation period of 95 °C for 15 min, followed by 35 cycles with the following settings: denaturation at 94 °C for 30 s, the annealing temperature of 65 °C for 90 s; extension at 72 °C for one minute. A final elongation step was performed at 72 °C for 10 min. PCRs were repeated up to three times per sample. For all 177 beetles, we used vertebrate-specific PCR primers to amplify portions of the mitochondrially encoded 12S rDNA (Ushio et al., 2017). For the Ecuadorian beetles (N = 135 beetles), we additionally used the mammal-specific primers 16smama1 (forward) and 16smama2 (reverse) (Taylor, 1996). For a subset of samples (N = 121 DNA extractions), we measured the DNA concentration using a ThermoScientific Nanodrop Lite Spectrophotometer (Thermo Fisher Scientific, Waltham, MA, USA). Of 177 beetles, 128 were amplified and sent to sequencing.

Proof of concept study

As a proof of concept, we offered a defined diet to Anoplotrupes stercorosus dung beetles consisting of a variety of mammals and amplified mammal DNA from the beetles’ guts. We then amplified the mammal DNA from the beetles’ guts. All 30 beetles were initially fed apples for five days to clear their guts. We then fed them with 10 grams of the following six dung types: tapir (Tapirus terrestris), fennec fox (Vulpes zerda), otter (Aonyx cinerea), porcupine (Hystrix cristata), macaque (Macaca nigra), and cow (Bos taurus). Beetles were fed for 24 h to ensure they had enough time to consume dung. Dung types were fed to five beetles each. Dung beetles were then euthanized in the freezer. The dung was contributed by a local zoo (Vivarium Darmstadt) and a local farm. We then applied our method as described above to identify dung beetle gut content. The amplified 12s rDNA fragment was then compared with a reference library using the Basic Local Alignment Search Tool (BLAST) to verify if it matched the consumed mammal dung (Altschul et al., 1990). Please note that we only used one species and a relatively low sample size for this proof-of-concept experiment, so caution should be taken in extrapolating the results to other dung beetle species.

Gut content fullness of beetles in pitfall traps after 8 h, 24 h, and 48 h

Pitfall trapping means that live beetles might empty their guts in the trap before being collected. At our German field site, we experimentally assessed how many beetles would still have full guts after pre-defined time periods. To approximate realistic gut fullness in pitfall traps, we set out 15 traps in the forest. The traps were baited with cow dung and randomly assigned to three different groups: five traps were emptied after 8 h, five traps were emptied after 24 h, and five traps were emptied after 48 h. Only A. stercorosus beetles were collected and immediately placed in 70% ethanol. In this proof-of-concept study, we solely used A. stercorosus to assess the level of gut fullness. Due to its larger size compared to Aphodius sticticus, the gut is easier to dissect reliably. In addition, they are also more readily available in Germany, and it is easier to identify the species in the field. The smaller Aphodius beetles are much harder to reliably identify. Beetle abdomens were then dissected to record gut fullness (at least 50% of the gut filled) or less than 50% filled. Data were analyzed with a Pearson’s chi-squared test.

Gut content after 48 h of starvation

A. stercorosus beetles (N = 20) were collected from the forest, transferred alive into different enclosures, fed cow dung for five days, and then starved for two days to survey the proportion of empty guts. After 48 h, the beetles were frozen and then dissected. Contents of guts (empty, not empty, full) were recorded together (Fig. S1). As a control, 20 beetles with ad libitum access to cow dung were frozen simultaneously to survey how many beetles had an empty gut when provided a continuous supply of food. Data were analyzed with a Person’s chi-squared test.

DNA sequence processing

For gut DNA analysis, we dissected 177 beetles from the two field sites belonging to 10 dung beetle species. Eight of the species were from Ecuador: Deltochilum sp.(8), Oxysternon conspicillatum (53), Canthon angustatus (58), Onthophagus sp. (8), Dichiotomius sp. (1), Canthidium sp.(3), Sulcophanaeus notis (1), and Scybalocanthon trimaculatus (2), as well as one unidentified species; two species were from Germany: A. stercorosus (19), and A. sticticus (23). PCR products of successful DNA amplifications, verified by gel electrophoresis and staining with ROTI® GelStain, were sent for Sanger sequencing to Macrogen Europe. Forward and reverse directions were sequenced for each amplicon. Post-processing was done using Codon Code Aligner 10.0.2 on macOS High Sierra. Low-quality base pairs (base pairs with a quality score lower than Phred 20) were clipped from the ends of the sequences. A Phred 20 quality score corresponds to 99% accuracy in a base call. At this stage, low-quality sequences were discarded (sequences with 50 or more base pairs with a score lower than Phred20) or sequences with a length of less than 50 base pairs. The resulting sequence length and quality scores were then recorded. Sequences were aligned, if possible, to create a consensus sequence. The resulting consensus sequences were then matched to reference sequences using the NCBI MegaBLAST search (Morgulis et al., 2008). We accepted the best match as our ID if the match was >90%, which is commonly used for 12s rDNA and 16s rDNA short sequences at the genus level for mammals (Hoffmann et al., 2017; Kocher et al., 2017; Drinkwater et al., 2019; Saenz-Agudelo et al., 2022). The best species match was recorded, along with the accession ID, percent identity, max score, and bit score. Mismatches between references and query sequences were often found in base pairs with a low-quality score. Lower matches might have partly arisen from DNA degradation as we analyzed DNA within digestive tracts. Due to this constraint and previously established protocols (Hoffmann et al., 2017; Kocher et al., 2017; Drinkwater et al., 2019; Saenz-Agudelo et al., 2022), we decided to identify the mammal species only to the level of the genus when there where multiple species within a genus otherwise we used the species name.

Data analysis

For the pitfall trap dataset, we ran a logistic regression with positive electrophoresis results as the dependent variable and gut fullness, collection method, PCR primers, DNA concentration, and beetle species as predictor variables. We then performed a dominance analysis (Budescu, 1993) to determine which of the predictor variables were the most important predictors of DNA amplification. A dominance analysis compares predictor variables in a pairwise fashion across all the subset models and generates a predictor hierarchy or the importance (dominance) of each predictor value. Higher values have a greater predictive power (Azen & Budescu, 2006; Lee & Dahinten, 2021).

We also constructed bipartite food networks (mammal dung–dung beetle) for each site to investigate the specificity of dung choice. For this, we counted the number of beetle individuals containing DNA of a given mammal genus. Our Ecuador data set includes 41 beetles whose gut content DNA successfully amplified and from which we successfully obtained sequence matches out of a total of 135 dissected beetles. The German network includes data from 21 beetles of the total 42 dissected beetles. Our ‘interaction frequency’ was the number of beetle individuals per species that contained a specific mammal DNA sequence. We calculated network specialization using the H2′ statistic for each plot (Blüthgen, Menzel & Blüthgen, 2006) as in Frank et al. (2018b). We compared H2′ values to randomized networks using the Patefield null model (see Blüthgen, Menzel & Blüthgen, 2006). We excluded sequences matching the bait (human in Ecuador and cow in Germany). Beetle species with no measured trophic interactions with mammal dung were dropped from the network analysis. We included humans as dung beetle interaction partners in our German network for two reasons. First, we observed a fair amount of human excrement at the German study site. Second, we showed that the newly developed washing protocol drastically decreased the detection of human contamination in the samples. Although contamination with human DNA cannot be entirely excluded, we consider it likely that most of the detected interactions with human dung at the German study site are accurate. When displaying the networks, we use species names if only one species of the genus is present in the study site, or we use spp. to indicate that more than one species is possible; however, we caution that using species distribution information in this way could make investigators overlook species with expanding ranges, or closely related genera could be misidentified. Finally, to estimate the diversity of mammal dung consumed by beetle species within each habitat type, we calculated the Shannon index for each beetle species and their associated mammal diversity.

Results

Proof of concept

We offered six dung types to a single dung beetle species. We verified the mammal genera for all six test dung types by matching DNA barcodes from beetle gut contents. The best DNA barcode match corresponded to the respective species for tapir, otter, porcupine, and cow (the top 10 best matches are the expected species), but macaque and fox have mixed species in the top 10 best matches. The 16s rDNA or 12s rDNA fragments for mammal identification are often used for genus level identification because this increases the accuracy of identification (Hoffmann et al., 2017; Kocher et al., 2017; Drinkwater et al., 2019; Saenz-Agudelo et al., 2022). Sequence quality was high, with a Phred score of 20 or higher from all base pairs. Data are available in the Supplemental Information.

Pitfall trap gut content fullness after defined hours

Among all beetles from traps emptied after eight hours, 75% had full guts (N = 15 total beetles, four beetles with empty guts and 11 beetles with full guts). After 24 h, 61% of the beetles had full guts (N = 44 total beetles, 17 beetles with empty guts and 27 beetles full guts), and 58% had full guts after 48 h (N = 24 total beetles, ten beetles with empty guts, and 14 beetles with full guts). Despite a decreasing proportion of full guts over sampling time, the three time points did not differ significantly in their proportion of full guts (Pearson’s chi-squared test: Chi2 = 0.95, df = 2, p = 0.621).

Gut fullness after 48 h of starvation

In a trial testing how many beetles could be expected to retain full guts after 48 h, we found that among the 20 beetles included in the starvation treatment, half the beetles (N = 10) had empty guts, nine guts were half full, and only one gut was full. Among the 20 beetles where food was available ad libitum for 48 h, five beetles had empty guts, five beetles had half full guts, and ten beetles had full guts. The proportion of full guts is thus lower in the starvation treatment than in the fed treatment. Accordingly, the two treatments differed (Pearson’s chi-squared test: Chi2 = 10.17, df = 2, p = 0.006).

Field collected dung beetles for gut content identification

Factors associated with successful DNA identification

Digestive tract fullness and DNA concentrations were the best predictors of measurable DNA amplification (Table 1). From the 177 beetles dissected across the two sites, we obtained 137 unique sequences of the rDNA gene fragments 12s and 16s. Success rates were higher for hand-collected dung beetles, 66% (27/41) than for dung beetles caught in pitfall traps, 49% (66/136) (Table S1).

Table 1 Results of dominance analysis.

The dominance analysis suggests that digestive tract fullness and DNA concentrations are the best predictors of measurable DNA amplification. The r2m approximates the importance of different predictor variables for a positive result from the gel electrophoresis within the model and is the average contribution of each of the five variables.

Variable	Dominance statistic	
Beetle species	0.051	
Collection method	0.009	
DNA concentration ng/ul	0.291	
Gut fullness	0.341	
PCR primers	0.009	
Notes.

Bolded values are the best predictors of DNA amplification.

The majority of dung beetles collected in pitfall traps had visibly full stomachs (31 out of 51), while fewer beetles had half-full or empty stomachs (20 out of 51). The success rate for amplifying mammal DNA dropped from 74% for beetles with full stomachs to 22% for empty stomachs (Table 2). Additionally, DNA concentration tended to be higher in cases of successful DNA amplification (median = 73 ng/µL mean = 86 ng/µL for positive result) versus those where DNA amplification failed (median = 34 ng/µL, mean = 46 ng/µL for negative results).

MegaBLAST search results

We amplified rDNA in 128 beetles out of 177 beetle guts. Of these 128 beetles, we only used 93 for the network analysis, either because the result from BLAST did not match a mammal (e.g., match to bacteria Klebsiella pneumoniae), resulted in an NA (matched nothing in the database), or the best match was lower than 90% which we considered too low to make inferences about genera or species. With a mean percent sequence identity of 96% at the species level for the 12s rDNA primers and 95% at the species level for the 16s rDNA primers, the best MegaBLAST matches were almost identical for the two primer combinations. These sequence matches were considered too low to make mammal species-level identifications but high enough to infer genus-level identities.

Table 2 Number of beetles with different gut fullness out of 51 specimens collected in total.

Showing that visible gut fullness was a good indicator of successful DNA amplification within our study. Data derive from eight dung beetle species: Deltochilum sp. (n = 5), O. conspicullatum (n = 21), C. anagustatus (n = 2), Onthophagus sp. (n = 1), Anoplotrupes stercorosus (n = 19), Dichiotomius sp. (n = 1), S. notis (n = 1).

	Full	Half Full	Empty	
Total Number of Beetles	31	11	9	
Number of Amplified DNA sequences	21	6	2	
Success rate	74%	55%	22%	

The 12s rDNA fragment had a higher chance of matching a species in the reference database (94% matching a mammal species present in study sites), while 55% of sequences of the 16s rDNA fragments did not match a mammal species present at our study site. However, matches at the genus level were better for 16s rDNA. There are three monkey species Ateles fusciceps, Alouatta palliata, and Cebus capucinus, at the field site in Ecuador (Tirira, 2017). Therefore, it is reasonable to assume that a genus level match can be used to identify the species in monkeys. However, that will be harder for more species rich orders of mammals such as Chiroptera and Rodentia. In the German network, there are two genera from Rodentia with multiple species, i.e., Apodemus and Myodes. In Ecuador, one genus has multiple species, i.e., Caloromys. Due to the possible lack of reference sequences with 16s rDNA, we instead focused on the 12s rDNA locus for network analysis. With respect to genus, the two primers produced the same result, suggesting some consistency within the results (Table S2–Table S4). The 12s rDNA primers provided one additional mammal species when compared with the 16s rDNA primers. Additionally, the storage method may be important for successful DNA amplification and subsequent sequence matching. The beetles from Ecuador stored in EtOH and frozen, were less likely to amplify and pass all quality control steps (41/135 beetles; 30%) than our German or frozen only beetles (21/42 beetles; 50%). We caution that these data are highly confounded with beetle species, PCR primer, temperature, and transport time from the field to the lab and, thus, should not be used to justify one method over the other.

Networks

With the dung beetles from Ecuador, we used both 16s and 12s rDNA primers. The network for the 16s rDNA primers is much less complete than the network generated using the 12s rDNA primers (Figs. S2 and S3). The 16s rDNA Ecuador network has an H2′ = 1, and the 12s rDNA network has an H2′ = 0.66. Both are significantly different than the null model (both p < 0.001). However, after the data from both primers were combined, the level of network specificity dropped to H2′ = 0.49, also significantly different from the null model (p < 0.001) (Fig. 1). The mammal dung–dung beetle network from Germany showed a similar moderate degree of specificity (H2′ = 0.41, p = 0.005; Fig. 2). We detected dung of 14 mammal genera from 10 dung beetle species (Tables S2, S3 and S4). These included top predators (e.g., P. onca (jaguar)), herbivores (e.g., Capreolus capreolus (deer)), and omnivores (e.g., Caluromys sp. (opossum)) (Tables S5 and Table S6). Per beetle species, we detected between one and six mammal genera (Tables 3 and 4).

Figure 1 Dung beetle mammal dung network Ecuador 12s and 16s Primers.

A bipartite network visualizing the links between dung beetle species and types of mammal DNA within their gut contents. The network consists of five dung beetle morphospecies and seven mammal genera. Sequences corresponding to the bait (human) were excluded. Line widths are proportional to the strength of the association, with thicker lines representing stronger observed links.

Figure 2 Dung beetle mammal dung network Germany 12s primers.

A bipartite network visualizing the links between dung beetle species and types of mammal DNA within their gut contents. The network consists of two dung beetle species and seven vertebrate genera. Sequences corresponding to the bait (cow) were excluded. Line widths are proportional to the strength of the association, with thicker lines representing stronger observed links. *included one bird species.

Table 3 Ecuadorian forest network.

Table summarizing the dung beetles species statistics including individual beetles per species, number of samples sequenced, match mammal species richness, and diversity.

Beetle species	N Beetles	DNA amplification with PCR	Mammal richness	e H′	
Canthon angustatus	29	23	5	3.11	
Canthidium sp.	3	0	NA	NA	
Deltochilum sp.	6	4	4	1.89	
Dichiotomius sp.	1	0	NA	NA	
Oxysternon conspicillatum	36	9	4	2.60	
Onthophagus sp.	8	3	3	3.00	
Sulcophaneus notis	1	0	NA	NA	
Scybalocanthon trimaculatus	2	2	1	1.00	

Table 4 German forest network.

Table summarizing the dung beetles species statistics including individual beetles per species, number of samples sequenced, match mammal species richness, and diversity.

Beetle species	N beetles	Positive PCR	Mammal richness	e H′	
Anoplotrupes stercorosus	19	12	6	5.32	
Aphodius sticticus	23	12	2a	1.38	
Notes.

a One bird species included here, all other samples were mammals.

Discussion

We investigated resource specialization in dung beetles by constructing bipartite interaction networks based on the barcoding of mammal dung derived from the beetles’ digestive tracts. Our method is broadly applicable, both geographically and phylogenetically. In the present work, we analyzed a tropical and a temperate forest community of dung beetles, including species of all of the three prominent dung beetle taxa (Scarabaeinae, Aphodiinae, and Geotrupidae). The greatest degree of DNA amplification success was found in beetles with visibly full guts. Hence, future work should best consider extracting DNA preferentially from beetles with full guts, which has the potential to cut down on expenses and time investment.

Previous work on dung beetle diets mostly produced ‘experimental’ (artificial) mammal dung–dung beetle networks by using laid-out dung or providing direct observations on natural dung sources (Young, 1981; Hanski & Cambefort, 1991; Frank et al., 2018a; Frank et al., 2018b; Raine & Slade, 2019). We see the present work as a step forward as DNA gut barcoding allowed us to provide more representative and natural networks. For instance, networks using laid out dung were highly generalized (mean ± sd H2′ = 0.23 ± 0.17 in 116 datasets, Frank et al., 2018b), while our two DNA-based networks showed a much higher level of specialization (H2′ = 0.41 and 0.49). Partly, this is because the studies included in the Frank et al. (2018a), Frank et al. (2018b) meta-analysis and others were often limited by dung access and often used dung from available domestic animals or animals from local zoos rather than naturally occurring dung in a habitat (Martín-Piera & Lobo, 1996; Errouissi et al., 2004; Korasaki et al., 2013; Frank et al., 2018a; Raine et al., 2018; Ong, Slade & Lim, 2020). Experimentally laying out dung could change environmental variables that are important for both attractiveness (dung volume) and natural encounter rates on the landscape (activity windows). Dung attractiveness is driven by dung volume in pitfall traps. The volume is often standardized across species instead of using naturally occurring defecates that vary in size (Errouissi et al., 2004). Dung beetles are also active at different times of day and, in many ecosystems, extremely efficient dung removers. This means mammal dung from nocturnal mammals is more likely to be encountered and consumed by nocturnal dung beetles than by diurnal dung beetles. The reverse also stands. Using only direct observations of dung beetles and mammal dung is open to a lot of observer bias, including observations of the most obvious dung types (e.g., elephant), or diurnal interactions over nocturnal interactions (Hanski & Cambefort, 1991; Scholtz, Davis & Kryger, 2009). The DNA-based method applied here minimizes these biases and helps to provide a more representative and detailed image of mammal dung–dung beetle trophic interactions. For example, our tropical interaction network suggests that nocturnal dung beetles are more likely to consume the dung of nocturnal mammals, e.g. P. onca and Dasypus novemcinctus. Furthermore, in the German temperate forest, we see less overlap in dung beetle diets than would be expected by random dung choice, under the expectation that dung beetles are generalist dung consumers (Hanski & Cambefort, 1991; Nichols et al., 2009; Frank et al., 2018a). Rodents were more dominant in the German mammal dung–dung beetle network than expected based on previous conclusions from the research of mammal dung–dung beetle networks (Hanski & Cambefort, 1991; Nichols et al., 2008; Frank et al., 2018a; Raine & Slade, 2019; Chiew et al., 2022). Previous research suggested that German networks are dominated by wild boar and deer, among other medium and large-bodied mammals, while rodents were mostly missing. However, a trend like that observed in the data might indicate excessive hunting or loss of large mammals like that suggested by Nichols et al. (2009). Both the present study and the one of Kerley et al. (2018) suggest that rodents have been largely overlooked using traditional methods to study dung beetle diets.

Despite the many advantages of the DNA-based approach, there are also some limitations. First, these networks measure adult diets. While there is much overlap between adult and larval diets, there are some suggestions that adults might provide different dung to their young than they consume themself (Byrne, Watkins & Bouwer, 2013; Shukla et al., 2016; Kerley et al., 2018). However, there is more evidence to the contrary, suggesting instead that adults and larvae both consume the same dung. Larvae still have chewing mandibles and may be able to take advantage of the solid parts of dung (grass fibers and other undigested material), while the adults only consume the liquid parts (Halffter & Edmonds, 1982; Hanski & Cambefort, 1991; Byrne, Watkins & Bouwer, 2013; Shukla et al., 2016). Second, some of the detected gut DNA may come from carrion instead of dung. However, the amount of available carrion and the integrity of the DNA is probably much lower than that of dung, particularly for older carrion (Itani et al., 2011; Yang et al., 2017). In addition, our results match that of the South African study (Kerley et al., 2018), where carrion is not considered part of a dung beetle‘s diet. Third, our method, like all pitfall trapping sampling methods, is likely to miss attracting dung beetles with highly specialized diets such as sloth dung (Young, 1981), snail mucus (Vaz-de Mello, 2007), or millipedes (Schmitt, Krell & Linsenmair, 2004). Fourth, we focused on the amplification of mammal DNA, which does not provide a complete picture of the beetles’ niche breadth. This may, in fact include plant material (Halffter & Halffter, 2009), other arthropods (Schmitt, Krell & Linsenmair, 2004; Silva, Vaz-de Mello & Barclay, 2018; Giménez Gómez et al., 2021), and snail mucus (Vaz-de Mello, 2007). By pointing out these deficiencies, we hope to inspire future research to tackle the method’s limitations, for example, using a broader set of bait attractants and/or analyzing gut DNA of a broader phylogenetic spectrum.

Overall, we see great potential in the analysis of dung beetle gut contents for various research areas. One noteworthy aspect of gut content DNA analysis is the potential to use it as mammal monitoring (see also Drinkwater et al., 2021). Like carrion feeding flies (Srivathsan et al., 2022), dung beetles can serve as ‘mammal samplers,’ and they could potentially constitute a more cost-effective and complete method than the traditional and widely used camera trapping (Drinkwater et al., 2021). The method is both spatially and temporally informative regarding mammal presence because the DNA degrades quickly, and dung beetles do not usually travel great distances within one day (Peck & Forsyth, 1982; Roslin et al., 2009; Silva & Hernández, 2015). By covering a broad dung beetle phylogenetic diversity and distinct geographic areas, the result of the present work suggests that gut barcoding can be broadly applied as mammal detector across distinct ecosystems. Further, the method could be broadly applied to the study of mammal dung–dung beetle networks, for example, to study dung beetle resource specialization in various habitats. Local scale changes, such as disturbance in natural habitats due to human activities, could be examined in terms of biodiversity and interaction shifts.

Conclusions

This study presented a gut DNA-based method to uncover mammal dung–dung beetle trophic networks, which will hopefully provide many new insights into these ubiquitous interaction networks. To maximize the success of future studies, we recommend that beetles should not be kept alive for more than 24 h. The recommendation is a conservative one but based on the results of the 48-hour starvation trial of two days without food significantly reduced gut fullness. Fuller guts should have more target DNA. This was also found to be a good predictor of a successful PCR amplification. Thus ensuring that beetles have limited time to void their guts is an important factor to consider. We also recommend washing the beetles and performing the dissections in a sterile environment to reduce human contamination, which can swamp target DNA. Finally, checking the gut contents before DNA extraction should markedly improve the success rate of future studies. Using DNA analyses of dung beetle guts could greatly improve our understanding of dung beetle biology within ecosystems and potentially provide an additional tool for biodiversity monitoring.

Supplemental Information

Table S1 Pitfall Trap collected beetles Ecuador Forest

8 pitfall traps

Table S2 Ecuadorian mammal list by primer from 8 pitfall traps, and 16 different hand collections

Table S3 From 18 paired sample results for 12s rDNA Primers

Table S4 From 18 paired sample results for 16s rDNA Primers

Table S5 List of mammals from the German forest from 7 pitfall traps using only the MiMammal-U primers (Ushio et al., 2017)

Table S6 Hand collected beetles Ecuador Forest 16 beetle hand collections

Supplemental Information 7 Sample data

Supplemental Information 8 Blast Results for Sequences

Supplemental Information 9 Gut Fullness in German Pit Fall Traps

Figure S1 Image of the categories of gut content filling for dissected dung beetles

Light brown indicates the maximum area with visible gut contents for the illustrated category, dark brown indicates the minimum area with visible gut contents for the illustrated category, and gray indicates no visible content (A) full—more than half the length of the intestines is full; (B) half-full –half or less than half the length of the intestine is full; (C) empty—there is no visible content in the intestine.

Figure S2 Ecuador Network from the16s Primers

A bipartite network visualizing the links between dung beetle species and types of mammal DNA within their gut contents. The network consists of three dung beetle morphospecies and three mammal genera. Sequences corresponding to the bait (human) were excluded. Line widths are proportional to the strength of the association, with thicker lines representing stronger observed links.

Figure S3 Ecuador Network from the 12s Primers

A bipartite network visualizing the links between dung beetle species and types of mammal DNA within their gut contents. The network consists of five dung beetle morphospecies and seven mammal genera. Sequences corresponding to the bait (human) were excluded. Line widths are proportional to the strength of the association, with thicker lines representing stronger observed links.

Supplemental Information 13 Starvation Trail

Supplemental Information 14 R scripts and Report of Proof of Concept

We would like to thank Sinsoma GmbH for their help in developing the method. Christoph Merkel for his assistance in the lab. David Donoso for his assistance with permits, Reserva Canandé, and Tesoro Escondido for support in the field. Finally, assistance in the field provided by Adriana Argoti Avila, Bryan Xavier Tamayo Zambrano, Jorge Alipio Zambrano Velez, José Amado De la Cruz Chávez, José Roberto de la Cruz Loor, Alcides Agustín Zambrano Velez, José Manuel Añapa Añapa, Ronaldo Mesías, Vanessa Moreira, Daniel Velázquez, Citlalli Morelos-Juarez and Yadira Giler.

Additional Information and Declarations

Competing Interests

Author Contributions

Field Study Permissions

DNA Deposition

Data Availability

The authors declare there are no competing interests.

Karen M. Pedersen conceived and designed the experiments, performed the experiments, analyzed the data, prepared figures and/or tables, authored or reviewed drafts of the article, and approved the final draft.

Christoph von Beeren conceived and designed the experiments, authored or reviewed drafts of the article, and approved the final draft.

Arianna Oggioni conceived and designed the experiments, performed the experiments, analyzed the data, authored or reviewed drafts of the article, and approved the final draft.

Nico Blüthgen conceived and designed the experiments, analyzed the data, authored or reviewed drafts of the article, and approved the final draft.

The following information was supplied relating to field study approvals (i.e., approving body and any reference numbers):

Escuela Politecnica Nacional, Contrato Marco MAE-DNB-CM-2016-0068 (Collection Permit).

The following information was supplied regarding the deposition of DNA sequences:

The raw sequence reads are available at Bioproject: PRJNA907546.

The following information was supplied regarding data availability:

The data sets, including the sample information and R scripts, are available in the Supplemental Files.

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
