# Peer review of "Mammal dung–dung beetle trophic networks: an improved method based on gut-content DNA"

_PeerJ, doi:10.7717/peerj.16627_

## Round 0.1 · original submission · Minor Revisions

Dear authors,

Please read the reviewers comments carefully and submit a revised manuscript addressing the revisions they suggest.

We are sorry the review process has taken so long. A large number of reviewers were invited before finding two who accepted the invitation.

·

Basic reporting

The paper is well-structured, clear, and understandable. Sufficient background and context are provided in the Introduction. The authors could consider including these additional references:
Srivathsan, A., Loh, R. K., Ong, E. J., Lee, L., Ang, Y., Kutty, S. N., & Meier, R. (2022). Network analysis with either Illumina or MinION reveals that detecting vertebrate species requires metabarcoding of iDNA from a diverse fly community. Molecular Ecology. https://onlinelibrary.wiley.com/doi/10.1111/mec.16767
Chiew, L. Y., Hackett, T. D., Brodie, J. F., Teoh, S. W., Burslem, D. F., Reynolds, G., ... & Slade, E. M. (2022). Tropical forest dung beetle–mammal dung interaction networks remain similar across an environmental disturbance gradient. Journal of Animal Ecology, 91(3), 604-617. https://besjournals.onlinelibrary.wiley.com/doi/abs/10.1111/1365-2656.13655

Experimental design

Based on the methods, the authors have conducted multiple proof-of-concept experiments (lab feeding trials, gut contents from wild caught beetles after pre-defined time periods). It would be clearer if the authors could highlight these experiments in the Introduction (with hypotheses). The methods described by the authors are generally clear, and I have a couple of specific queries to improve the clarity:
Lines 130 to 131: Please provide the number of pitfall traps set in Germany and Ecuador respectively. Kindly clarify why different dung types were used for sampling in both sites too.
Lines 136 to 137: Was there a difference in DNA quality between the two DNA preservation methods (i.e. ethanol and frozen VS frozen only)?
Lines 164 to 167: I feel that it will be very useful to provide a visual representation of these gut fullness categories, especially for future studies that seek to apply the authors’ methods.
Lines 171 to 180 (DNA extraction and amplification): Please indicate the elution volumes used for the extraction. Kindly provide more details on the PCR recipe and conditions (may refer to the methods in https://onlinelibrary.wiley.com/doi/10.1111/mec.16767 as a detailed example). Have the authors considered using qPCR (e.g. https://peerj.com/articles/11897/) to determine the actual DNA concentration? Kindly clarify why 121 samples were quantified and not all 177 of them.
Lines 189 to 190: Please state the duration where the beetles fed on the dung and clarify whether they were removed/killed immediately after.
Lines 201 to 202 and Line 207: Please clarify as to why only one species of beetles was used for the experiments (potential limitation?) as the gut fullness may differ across beetle species.
Line 217: Were all 177 samples successfully amplified?
Lines 235 to 241: Please state the type of analyses performed with the main study and proof-of-concept experiments for better clarity (e.g. chi-sq test for gut fullness; logistic regression/dominance analysis for main study?)
Line 243: Are the networks based on both 12S and 16S primers? Have the authors considered constructing networks for each of the primers (12S VS 16S)?

Validity of the findings

Based on the results, the authors have achieved the study’s main goal of developing and applying their technique of using dung beetle gut contents to construct trophic networks. For the proof-of-concept studies, the results are largely preliminary due to the low sample size and use of only one species. This is understandable and the results are still useful in informing future studies. The authors could acknowledge this as a study limitation.
Figure 2: One bird species was detected – was this from the 12S primers? Could this be an artefact as the 12S primers were supposed to be specific to mammals (according to Ushio et al 2017)? Would the authors be able to deduce if the Homo sequences were due to contamination or actual feeding?

Additional comments

This paper presents a very interesting approach of sampling dung beetle gut contents and applying DNA metabarcoding techniques to develop novel dung beetle mammal trophic networks based on direct interactions between dung beetles and the dung of mammals that they fed on.
The methods developed by the authors are very useful for future studies that seek to construct such networks in other regions and aids in the understanding of ecological interactions and services provided by dung beetles.

Reviewer 2 ·

Basic reporting

This manuscript is written clearly and professionally, with only a handful of small sentence structure points. The introduction and discussion are clear and there is excellent evidence of wide research, which is well used to support the test. The wider field of dung beetle trophic networks is well described and this allows the authors to set a strong rationale for the study. The article is generally well structured to the requirements of the PeerJ author guidelines. It should be notes that there are several separate case studies within this manuscript, and this makes some aspects of the methods and results more challenging to follow. The data availability statement indicates that data will be shared if the work is accepted.

Experimental design

The experimental design appears largely valid. The trapping of dung beetles is well explained and while it would be nice to have a bit more detail on the sampling periods and numbers of beetles, the key areas in terms of analysis are well explained. Ethics should be acknowledged however, especially with sampling of live animals. While beetles are not always afforded welfare, there should be some consideration of any ethics reviews that took place prior to study commencement. Similarly, consideration of human biowaste and author PPE should be acknowledged briefly.
This said, this research is clearly novel and fits a gap within the existing literature. There are some well justified arguments for this research - though future directions could be considered further.

Validity of the findings

Generally, the findings that have been claimed are valid though there does need to be a little more clarity regarding the stating of significance for gut filling as this doesn't seem to match the results section. This said, the selection of statistical tests is well justified and appropriate, and the figures are supportive of the wider text. Inclusion of the raw data would be useful from a reviewer standpoint. Make sure the points in the conclusion are fact checked with regards to significance.
As there is a prof of concept aspect to this study, future research avenues could be discussed in a little more depth. This would help to encourage relevant future research.

Additional comments

Overall, this is an interesting and useful manuscript. A little more clarity on aspects of the methods, and clarity on significance stated in methods, would help to improve this manuscript further.

Annotated reviews are not available for download in order to protect the identity of reviewers who chose to remain anonymous.

---

## Round 0.2 · accepted · Accept

Thank you for addressing the reviewers' comments completely. I look forward to seeing this published.

·

Basic reporting

No comment.

Experimental design

My only concern is the use of Sanger sequencing vs NGS, which the latter allows for greater sequencing depth and detection of more sequences (e.g. rare vertebrate species). It will be ideal if the authors could justify their use of Sanger sequencing (e.g. not expecting high number of targets in study system?) or highlight this as a limitation, given that this study is also a proof of concept.

Validity of the findings

No comment.

Additional comments

Overall, this is a very interesting study and I commend the authors for thoroughly addressing the reviewers' concerns raised.

Reviewer 2 ·

Basic reporting

The revisions to this document have resulted in a clearer manuscript in terms of readability. There has been additional references added within the text, and they have been used in such a way that they add credibility to the work. The aim of the work is clear.

Experimental design

Ethical information has now been added into the manuscript. The concerns relating to biohazardous material (human waste products) have now been explained more clear. The authors have addressed the initial concerns on the manuscript well.

Validity of the findings

This is an useful and meaningful manuscript that fills in a gap in the wider literature. The further information within the methods and the developments to the results section has resulted in a clearer manuscript overall.